# Attitudes towards COVID Vaccine and Vaccine Hesitancy in Dermatology: A Narrative Review

**DOI:** 10.3390/vaccines11081365

**Published:** 2023-08-15

**Authors:** Woo Chiao Tay, Anthony Bewley, Julia-Tatjana Maul, Hazel H. Oon

**Affiliations:** 1National Skin Centre, 1 Mandalay Road, Singapore 308205, Singapore; 2Department of Dermatology, Barts Health NHS Trust, London E11 1NR, UK; 3Queen Mary University, London E1 4NS, UK; 4Department of Dermatology and Venereology, University Hospital of Zurich, 8091 Zurich, Switzerland; 5Faculty of Medicine, University of Zurich, 8006 Zurich, Switzerland; 6National Skin Centre, Skin Research Institute of Singapore, 1 Mandalay Road, Singapore 308205, Singapore

**Keywords:** mental health, psychodermatology, social isolation, skin health, vaccines safety, vaccine hesitancy

## Abstract

Vaccine hesitancy has been a contentious issue even before the pandemic. The COVID-19 crisis has further amplified vaccine hesitancy, with worries about adverse effects, cultural and religious beliefs, and misinformation on social media. In dermatology, patients with pre-existing skin conditions may have specific concerns about the impact of the vaccine on their skin health. Factors such as cutaneous reactions, potential flares of underlying conditions, and fears of psoriasis worsening post-vaccination contribute to vaccine hesitancy. Healthcare professionals, including dermatologists, play a crucial role in addressing vaccine hesitancy by providing accurate information, addressing concerns, and understanding the psychological impact on patients. The concept of vaccine fatigue is also explored, noting the challenges in sustaining vaccine acceptance over time, especially with regards to booster vaccinations. Overcoming vaccine hesitancy requires trust-building, effective communication strategies, and collaboration between healthcare workers and non-healthcare individuals to combat misinformation. By recognizing and addressing psychological factors, dermatologists can increase vaccine acceptance and improve public health efforts.

## 1. Introduction

Vaccines have always sparked debates about their safety [1,2]. Vaccine hesitancy is defined as the delay in acceptance or refusal of vaccination despite the availability of vaccination services [3]. In 2019, the World Health Organization (WHO) recognized vaccine hesitancy as one of the top 10 global health threats, even prior to the COVID-19 pandemic [4]. However, the pandemic has brought vaccine hesitancy, particularly among patients with skin conditions, to the forefront [5].

Vaccination, as a public health intervention, is a successful preventive strategy that has saved multiple lives through history [6]. However, this does not mean that everyone is in favour of it [2]. Even before COVID-19 vaccines were available, vaccination has been a subject of controversy, with differing opinions among various groups [1,7,8,9]. The 1998 Lancet paper by Wakefield et al., which was eventually retracted, reported a perceived connection between the MMR vaccine and autism.

Understanding the reasons behind vaccine hesitancy and its implications for public health is crucial. Vaccine hesitancy is a multifaceted issue, even more so in individuals with pre-existing skin conditions. This review explores the key arguments surrounding vaccine hesitancy, from cultural and religious beliefs to the importance of education and communication, while also addressing strategies to overcome it.

This narrative review summarizes the recent literature on these topics. We mainly focus on articles published after December 2020, after the first COVID-19 vaccine (Pfizer-BioNTech mRNA vaccine) was approved for emergency use by the World Health Organisation, identified by the PubMed search in English. Keywords used in the PubMed searches were COVID-19 vaccine (or SARS-CoV-2 vaccines), attitudes towards vaccines, vaccine acceptance (or acceptability or willingness or hesitancy or refusal), dermatology (or dermatological conditions or cutaneous), and vaccine fatigue. The International Narrative Systematic Assessment (INSA) tool was used as a framework for assessing this narrative review (Table 1) [10].

## 2. Vaccine Utilization Prior to COVID-19 Pandemic

Many patients with skin conditions receive immunosuppressant therapy (traditional oral systemic agents or newer biologic agents), making it essential for them to maintain an up-to-date vaccination schedule. However, even prior to the COVID-19 pandemic, dermatologists have not consistently recommended and administered vaccines as part of routine patient care [10]. Several studies also indicated that dermatology patients on immunosuppressants did not display a higher likelihood of receiving vaccinations [11,12,13,14].

A cross-sectional study conducted in France with 68 patients with psoriasis on immunosuppressants (cyclosporine, methotrexate, etanercept, infliximab, adalimumab, or ustekinumab) revealed that only one patient was up to date with all recommended vaccinations [11]. The most common reasons for not updating the immunization schedule were the absence of any notification or proposal from the patient’s doctor, and oversight. This underscores the need to emphasize awareness among both patients and healthcare workers regarding vaccination recommendations.

Another cohort study conducted in the US among psoriasis patients yielded similar findings, showing that patients on systemic agents or biologicals did not exhibit an increased likelihood of receiving the influenza vaccine [12]. In this study, the psoriasis patients were younger and had lower rates of certain chronic diseases, such as chronic kidney disease, chronic obstructive pulmonary disease, congestive heart failure, and diabetes, which might have contributed to a lower perception of risk of influenza infection and its complications. Psoriasis patients also had concerns about the safety of receiving vaccinations, particularly while they were receiving systemic therapies. Additionally, inadequate counselling from physicians was reported as a concern for patients with psoriasis, as well as the underlying flare of the underlying skin condition [15].

## 3. Dermatology Patients and Attitude towards COVID-19 Vaccination

While dermatologic patients generally displayed positive attitudes toward COVID-19 vaccination, many factors can have an influence on their attitudes. A single-institution study conducted on 713 patients with psoriasis at a dermatology centre in Greece revealed that patients with psoriasis were 32% more willing to receive the COVID-19 vaccine compared to the control group (odds ratio 1.32) [16]. Among patients with psoriasis, individuals with psoriatic arthropathy were nearly 20% more likely to undergo COVID-19 vaccination (odds ratio 1.18) after adjusting for age, sex, and type of treatment. Younger patients with psoriasis were more inclined to be vaccinated compared to individuals without psoriasis.

Another survey involving nearly 300 dermatology patients in Turkey categorized respondents into two groups: those below 40 years old and those above 40 years old [17]. The older group exhibited significantly higher levels of anxiety compared to the younger group (*p* = 0.017). While approximately 60% of cases in the older group were willing to be vaccinated, 40% of younger participants expressed uncertainty about vaccination (*p* < 0.001). The most requested vaccine types were inactivated vaccines for the elderly and mRNA vaccines for the younger group (*p* < 0.001). Weak statistically significant positive correlations were observed for age, chronic medication use, and the presence of severe COVID-19 cases in the environment.

Rheumatology and dermatology are similar in that many conditions have an autoimmune basis in its pathogenesis. Similar concerns about vaccine safety and the potential exacerbation of underlying inflammatory conditions have been observed among patients with myositis, rheumatic diseases, including systemic lupus erythematosus, and rheumatoid arthritis [18,19,20].

## 4. Factors Affecting COVID-19 Vaccine Uptake in Dermatology Patients

The COVID-19 pandemic has significantly amplified the discussion on vaccine safety and vaccine hesitancy, particularly among dermatology patients [21]. COVID-19 vaccines are available in four main forms: messenger RNA (mRNA) vaccines, viral vector vaccines, protein subunit vaccines, and inactivated virus vaccines. There is a myriad of reasons behind vaccine hesitancy, and they are often not isolated. MacDonald and the WHO Strategic Advisory Group of Experts on Immunization (SAGE) Working Group on Vaccine Hesitancy summarized these reasons into three main factors: confidence, complacency, and convenience [7].

(Lack of) Confidence: Some individuals worry about the quality of the production process, potential side effects, and the decision-making process of policymakers.Compliance: Some do not perceive the need for vaccination. This can be due to a lower perceived risk of COVID-19 infection, or complications from the COVID-19 infection.Convenience: Other factors beyond medical reasons, such as a preference for natural or alternative remedies, political inclinations, and extreme religious beliefs, can all contribute to an individual’s decision-making process. Reduced geographical and financial accessibility may make vaccines less appealing to certain individuals.

### 4.1. Knowledge of Vaccine Production Process and Side Effects, and the Impact of Social Media Infodemic

Some people hesitate to vaccinate due to concerns about the development and testing of vaccines, or a belief that natural immunity is superior to immunity provided by vaccines [22,23,24]. The vaccine development process typically undergoes years of preclinical evaluation and three distinct clinical stages before validation [25]. However, under extraordinary circumstances like pandemics, the development process may be expedited to reduce infection-related morbidity and mortality [26]. However, it was this rapid process that led to confusion and increased public concern regarding the efficacy and safety of newly developed vaccines. This scepticism may stem from a lack of trust in the government or pharmaceutical companies, or a belief that the vaccine has not undergone thorough testing [22,23,27]. Some individuals worry about reports suggesting that the COVID-19 virus itself was genetically engineered by governments.

Parents of young children also expressed concerns about the safety of vaccines. Parents of young children also raised concerns about the use of thimerosal, a mercury-based preservative, in vaccines [28]. A systematic review by Khan et al. examined 108 studies on vaccine hesitancy and reported that the most common barriers to childhood vaccination included mothers with lower education levels, financial instability, low confidence in new vaccines, and exposure to unmonitored social media platforms [29]. However, the same systematic review highlighted that measures such as providing information by healthcare professionals could improve vaccine uptake.

Despite the efforts of various governments to promote vaccine safety, a significant amount of misinformation and exaggeration of cutaneous adverse events associated with vaccines (e.g., “COVID arm”) continues to circulate on social media [30]. People believe that vaccinations can cause certain diseases or lead to long-lasting health problems [31]. In this era of booming information technology, the role of social media in disseminating medical information (or disinformation) and its impact on an individual’s medical choices cannot be underestimated [32,33,34,35].

Patients with dermatological conditions also seek information about COVID-19 vaccines and their potential impact on their skin health from online sources. A study analysing online social media posts about psoriasis medication interactions with COVID-19 vaccines found that out of 477 posts, 19 (4%) contained a negative sentiment, 232 (48.6%) were neutral, and 226 (47.4%) expressed a positive sentiment [36]. A significant number of posts (32.5%) expressed concerns about pausing or discontinuing medications prior to receiving the vaccine. Other common concerns included a fear of negative reactions (21.8%) and uncertainty about the ability to generate an efficient immune response to the vaccine while taking anti-psoriatic medications (19.9%).

The major driver amplifying anti-vaccine sentiments during the COVID-19 pandemic has been the social media “infodemic” [37,38,39]. Personalized algorithms on social media platforms select articles and content that align with users’ preferences, creating online echo chambers and artificially inflating the perceived public consensus on misinformation. In Croatia, participants who sought information on social networks (odds ratio 0.36), general internet/blogs forums (odds ratio 0.34), or from friends or acquaintances (odds ratio 0.66) had lower odds of being vaccinated [40].

A noteworthy randomized controlled trial conducted in the UK and US aimed to quantify the impact of exposure to online misinformation on vaccine attitudes [41]. Compared to exposure to factual information, recent misinformation induced a decline in vaccine intention of 6.2% in the UK and 6.4% in the US among those who had initially expressed definite acceptance of the vaccine. Different sociodemographic groups were also differentially affected by exposure to misinformation.

Efforts to combat “fake news” have been difficult. Only a small proportion of healthcare workers have actively stepped into the digital world to counter misinformation. Other healthcare workers, fearing retaliation from anonymous “online experts,” have chosen not to engage or correct false information on social media. Hernandez et al. coined this phenomenon as “Health Care Provider Social Media Hesitancy,” referring to the nonaction of healthcare workers in providing pro-vaccine and scientific information about vaccines on social media [37]. This nonaction allowed the misinformation online to grow unabated.

Spikes in vaccine hesitancy were also seen to coincide with the emergence of new information, policies, or newly reported vaccine risks [35]. Some factors contributing to this variability include a decline in public trust in experts, preference for alternative health approaches, political polarization, and belief-based extremism.

### 4.2. Impact on Skin Health or Worsening of Skin Conditions

In the medical literature, there is extensive documentation of cutaneous reactions following COVID-19 vaccinations. These reported reactions encompass a range of new onset inflammatory skin conditions, including psoriasis, eczema, immunobullous disorders like pemphigus vulgaris and bullous pemphigoid, lichen planus, urticaria, alopecia areata, morphea, pityriasis rosea, herpes zoster, chilblains, and vitiligo [42].

Compared to the general population, individuals with pre-existing dermatological conditions also expressed additional specific concerns with regard to the COVID-19 vaccines. This was especially seen in patients with psoriasis, urticaria, and previous local site reactions to vaccines, as they are at a higher risk of experiencing a recurrence of these conditions (either local koebnerisation or systemic flare) after receiving the COVID-19 vaccine [43]. A questionnaire administered to 707 patients from the International Pemphigus and Pemphigoid Foundation revealed that only 73.1% of patients were willing to accept the COVID-19 vaccine [44]. Respondents expressed concerns that the vaccine could trigger a flare-up or worsen the control of their underlying autoimmune bullous diseases.

Localized cutaneous reactions, such as urticarial and morbilliform eruptions, were also commonly reported after mRNA vaccines [43]. Infrequent cases of herpes zoster reactivation, dermatologic filler reactions, and immune thrombocytopenia were also reported, but mainly in high-risk patient groups [30].

Data from the global patient-reported PsoProtectMe survey indicated higher vaccine acceptance rates among patients with psoriasis. Only a minority of respondents (8%) reported vaccine hesitancy. Young psoriasis patients and those with negative experiences with healthcare and/or doctors were more likely to exhibit vaccine hesitancy. The most common reasons for hesitancy were concerns regarding the side effects of a new vaccine and potential worsening of psoriasis post-vaccination [45].

Among the 271 individuals from the Massachusetts General Hospital Vaccine Allergy Registry who experienced urticaria reactions following COVID-19 vaccination, 186 (69%) individuals expressed reluctance to receive future recommended doses of the COVID-19 vaccine, despite acknowledging the overall safety of these vaccines [46]. The hesitancy stemmed from a sense of personal protection from a poor reaction (urticaria), and not a new onset of distrust towards the COVID-19 vaccines. A scoping review of 60 articles (ACCORD) by Batac et al. identified the possibility of allergic reactions as a factor contributing to vaccine hesitancy in 22% of the studies [47]. This fear of allergic reactions was observed both in individuals living with allergies and those without a history of allergic diseases.

### 4.3. Experiences from Previous Healthcare Encounters

Individuals who had negative previous healthcare experiences were more likely to exhibit vaccine hesitancy. In the global patient-reported cross-sectional survey (PsoProtectMe), younger individuals with psoriasis, and those who had unfavourable encounters with healthcare or medical practitioners, demonstrated a greater tendency toward vaccine hesitancy [45].

In an international cross-sectional survey in 20 hidradenitis suppurativa patient support groups, participants who reported being dissatisfied with their hidradenitis suppurativa care were found to be more inclined towards hesitancy regarding COVID-19 vaccines and were also more likely to decline influenza vaccination [48].

### 4.4. Psychosocial Factors

Some individuals were hesitant to vaccinate due to strong cultural or religious beliefs against vaccination [49]. For example, in certain communities or religions, it is believed that vaccines can cause the spirits of loved ones to leave the bodies of those who were vaccinated.

In some instances, individuals believed that the COVID-19 virus was a bioweapon developed by the Chinese government [50]. Other conspiracy theories included notions that COVID-19 was not real or that it was an effort by the government to control society. Some even believed that the vaccine contained a tracking chip [24].

Some individuals may be hesitant to get vaccinated simply because they dislike needles or do not want to take the time to get vaccinated. Others may believe that they were not at risk of contracting the virus or developing complications [38].

Adults with eczema and psoriasis have also identified barriers such as limited access to appointments, timing issues, and travel requirements for vaccination as factors contributing to their decision not to receive the vaccine [51].

Lastly, some individuals experience “information paralysis,” where they become overwhelmed by the abundance of information about the COVID-19 virus and vaccines, leading them to refrain from making any decisions [24].

## 5. Improving Vaccine Acceptance and Role of Clinicians

Many strategies have been proposed to address vaccine hesitancy, although only a few have been evaluated for their impact [52,53,54]. A systematic review by Jarrett et al. examined 13 studies using social mobilization, mass media, communication-centred training for healthcare personnel, non-monetary incentives, and reminder/recall-based approaches. The results indicated that multicomponent and dialogue-based interventions were the most successful [8]. Messages must be clear and reduce the cognitive load of the reader to improve understanding and retention of the information [55].

A correlational study of 1095 subjects in Italy during the national vaccination campaign for the third dose showed that an individual’s intention to get vaccinated (or not) requires considering many sociopsychological factors, and trust in science plays a crucial role in predicting vaccination intention [56]. The authors called for additional strategies promoting healthy behaviour to improve vaccine acceptance.

Trust, or lack thereof, in medical professionals was a crucial factor in deciding whether an individual chooses to get vaccinated [37]. A survey of 2440 adults by Nowak et al. revealed that vaccine hesitancy was significantly associated with individuals having greater trust in friends and family than in medical professionals [57]. Involving patient associations decreases the use of a “top-down” approach in delivering medical messages and alleviates the negative pressure experienced by hesitant patients [58]. Patients feel that their concerns were heard and addressed, promoting trust in vaccination.

Apart from safeguarding the vaccinated person, vaccination serves as a vital public health measure by providing herd immunity, thus protecting those who cannot receive vaccines due to age, contraindications, or other medical reasons. It is crucial for healthcare workers to prioritize educating individuals at a higher risk of COVID-19 complications, such as those with hypertension, diabetes, or obesity, to overcome vaccine hesitancy [59]. Dermatology patients, particularly those undergoing CD20-depleting therapies like rituximab, high-dose corticosteroids, or other immunosuppressants, face increased risks compared to the general population [60].

From a population standpoint, adverse cutaneous side effects of vaccines are unfortunately inevitable. Individuals should be pre-emptively counselled about possible skin reactions to the COVID-19 vaccines. It should be emphasised that these reactions, which often occur within a few days following vaccination, are generally mild and self-limiting [43]. Individuals who have received counselling about the potential adverse reactions to the vaccines were less likely to feel negatively in the event such reactions occur.

The role of dermatologists in improving vaccine hesitancy has also been studied. In two dermatology practices in Texas, US, immunosuppressed patients who initially declined an influenza vaccine were provided dermatologist-led education on the benefits of immunization [61]. Dermatologists explored and addressed individual patients’ concerns regarding immunization. Influenza vaccination was then offered immediately following the dialogue. The study found that influenza vaccination was more likely in the intervention group compared to the comparison group (odds ratio 16.22).

The impact of these efforts, fortunately, was not lost, as evidenced by waning rates of vaccine hesitancy. A cohort study in the US compared attitudes towards vaccines and vaccine hesitancy at two timepoints and showed that nearly one-third (32%) of individuals who were initially hesitant became vaccinated at follow-up, and more than one-third (37%) transitioned from vaccine-hesitant to vaccine-willing [62].

In the future, the range and diversity of vaccine manufacturers and techniques are expected to expand. It becomes consequently imperative to thoroughly assess these upcoming methods [63]. Dermatologists should not confine themselves solely to medical and scientific approaches when confronting misinformation originating from religious, media, or governmental sources. It is crucial for them to collaborate with non-healthcare workers to effectively address and combat this damaging misinformation [38,49].

## 6. Healthcare Worker Attitudes toward Non-COVID-19 and COVID-19 Vaccines

Healthcare workers play a crucial role in the frontline battle against the virus, and their perspectives on vaccination are highly significant. Prior to the pandemic, a comprehensive review of the literature concerning healthcare workers and their attitudes towards vaccines revealed a predominant focus on influenza vaccination (84%), followed by hepatitis B, and pertussis [64]. Healthcare workers e.g., physicians [65], nurses [66], and even midwives [67] who have been vaccinated themselves were more likely to recommend vaccinations to their patients.

A survey conducted in the Chicago area involving 1974 responses reported that 99% of physicians were planning to be vaccinated, while only 82% of nurses expressed the same intention [68]. The study also found that healthcare workers who were Black (odds ratio 0.34) or Republican (odds ratio 0.54) were less likely to receive the COVID-19 vaccine. Another survey conducted among 2720 healthcare workers in southwestern Virginia revealed that 18% of respondents expressed vaccine hesitancy, with increased odds among participants who were Black, younger, had no high-risk household member, and had no prior personal experience with COVID-19 infection [69].

A study conducted during the first week of the COVID-19 vaccination campaign dedicated to Italian healthcare workers found that socio-demographic control variables such as age, gender, and seniority had little or no predictive power in vaccine recommendation. Instead, vaccine confidence, positive emotions, and internal locus of control were excellent predictors of vaccine recommendations by doctors [70]. Younger doctors, both in age and experience, demonstrated higher confidence in vaccines and recommended them more frequently.

In general, dermatologists showed acceptance towards the vaccine. However, a small proportion of dermatologists and dermatology trainees in Turkey expressed a firm unwillingness to receive vaccination [71]. The study also found a statistically significant, albeit weak, correlation between younger practitioners and a shorter duration of medical practice with a higher acceptance of the vaccines. It is worth noting that there may be confounders in this correlation, as the dermatology trainees in this study had a higher percentage of assignments to COVID-19 clinics and intensive care units, which put them at higher risk for contact with COVID-19 positive cases.

A questionnaire survey involving 184 dermatologists in Europe who care for patients with autoimmune bullous diseases advocated COVID-19 vaccination even during immunosuppressive treatment, with some restrictions for certain medications such as rituximab [72].

Estimates of vaccine hesitancy among healthcare workers were found to be similar to the general population [73]. This was an interesting finding, as one would expect healthcare workers to have greater acceptance towards the vaccine. It is essential to understand the reasons why some healthcare workers remain hesitant towards the COVID-19 vaccine and the implications this has for public health.

Healthcare workers, especially those working in acute hospitals with direct contact with COVID-19 patients and its complications, generally showed acceptance towards the COVID-19 vaccine. The top reasons cited for vaccine acceptance were to protect their family and friends and to protect themselves due to their occupational risk [74]. Healthcare workers who had contracted COVID-19 or had a close friend/family member contract COVID-19 were also more likely to accept vaccines [75].

However, some healthcare workers chose not to proceed with the vaccine due to concerns about adverse effects. It is important to consider that this perception may be influenced by selection bias, as healthcare workers on the front lines of the pandemic were more likely to witness patients with side effects and complications from the vaccine, creating an impression that overestimates the true incidence of vaccine complications and side effects.

Educating healthcare workers remains the most important factor in overcoming vaccine hesitancy among this group. This cannot be overemphasized. However, healthcare workers might also be hesitant to get vaccinated due to personal, cultural, or religious beliefs.

A large-scale online survey in Germany, which included 4500 participants, reported that healthcare workers expressed concerns about both the short-term (local reactions, allergic reactions) and long-term side effects (autoimmune reactions, neurological side effects, and unknown long-term effects) of the vaccines [76].

It is very important to recognize and respect different healthcare workers’ cultural and religious beliefs, as they come from diverse cultural and religious backgrounds [73,74,75,77,78]. These cultural and religious beliefs must be considered when educating healthcare workers about the vaccine, with appropriate medication to education tone and content.

It is important to highlight that the attitudes of healthcare workers are not fixed and can evolve over time, particularly with a deeper understanding of the vaccines. A study conducted in Lebanon demonstrated this phenomenon, as the influenza vaccination uptake rate among healthcare workers increased from 32.1% during the 2019–2020 cycle to 80.2% in the subsequent annual cycle [79]. This shift indicates that healthcare workers may become more receptive to vaccination as they gain more knowledge and experience with the vaccines.

Another study in Barcelona showed using two online surveys, conducted 6 months apart, of how the attitudes of the nurses changed with time [80]. In Singapore, a review of a database indicated that the rollout of the vaccination program improved vaccine hesitancy among healthcare workers, and they were less hesitant towards the COVID-19 booster than the first dose [81].

Reported strategies to improve COVID-19 vaccination uptake among healthcare workers in Southeast Asia include incentivizing vaccination efforts through measures such as allowing healthcare workers to choose their preferred vaccination brand, issuing immunity passports, providing time off from work, and offering subsidies for travel to vaccination centers. These initiatives aim to enhance vaccine acceptance and increase vaccination rates among healthcare workers in the region [82].

## 7. Vaccine Fatigue

The ongoing efforts to promote and advocate for COVID-19 vaccinations and boosters during the prolonged global fight against the pandemic have led to a phenomenon known as vaccine fatigue [83]. This phenomenon is not new and has been observed in the context of influenza vaccination, where suboptimal uptake has resulted in unnecessary deaths [84].

While COVID-19 vaccine hesitancy for primary vaccination seems to have stabilized, there has been a decline in acceptance of subsequent booster vaccinations [35,85,86]. This decline is concerning as COVID-19 vaccine immunity in the population is waning, and studies suggest that seasonal or regular booster vaccinations may be necessary, particularly for vulnerable groups such as the elderly and immunocompromised individuals [87,88,89]. However, modelling studies have demonstrated that administering well-timed vaccine boosters to all eligible individuals, rather than solely focusing on vulnerable populations, provides better protection and is cost-effective in reducing infections and hospitalizations [90]. Therefore, it is crucial for individuals who have received previous vaccinations to continue doing so to maintain herd immunity.

Stamm et al. evaluated determinants of COVID-19 vaccine fatigue by embedding two experiments in an online survey of 6357 participants in Austria and Italy [91]. The authors recommended customizing vaccination campaigns according to participants’ vaccination status. For the unvaccinated, messages emphasizing community spirit had a positive impact, while providing positive incentives like cash rewards or vouchers played a crucial role in influencing the decision-making process for those who had received one or two doses. Among those who had received three doses, their willingness to be vaccinated increased when adapted vaccines were offered, but concerns about costs and disagreements among medical professionals decreased their likelihood of getting vaccinated.

## 8. Strengths and Weaknesses

Strengths: This narrative review presents a comprehensive overview of vaccine hesitancy among individuals with dermatological conditions, covering both non-COVID-19 and COVID-19 vaccines, with a brief mention of vaccine fatigue. It also highlights the significance of healthcare workers, especially dermatologists, in influencing vaccine acceptance.

Weaknesses: Vaccine attitudes are dynamic and can change over time due to evolving virus characteristics and vaccine safety data. The review provides only a snapshot of the literature available at the time of writing, potentially missing new developments. Non-medical factors like cultural and personal beliefs, which greatly impact vaccine attitudes, are often underreported or not published in English literature, leading to a lack of specific examples that could be universally applied due to cultural variations worldwide.

## 9. Conclusions

Vaccine hesitancy among individuals with dermatological conditions poses challenges to public health efforts. The multifaceted nature of vaccine hesitancy necessitates a multipronged approach that simultaneously addresses concerns about safety, side effects, and misinformation. Healthcare workers, especially dermatologists, play a crucial role in promoting vaccine acceptance, as their attitudes significantly influence patient decisions. Dermatologists can provide tailored information to patients and address specific concerns related to skin health. Efforts to combat vaccine hesitancy should prioritize education, clear communication, and trust-building, considering cultural and religious beliefs. As the fight against the COVID-19 pandemic continues and the need for booster vaccinations arises, sustaining vaccine acceptance and addressing vaccine fatigue become increasingly important. By implementing effective strategies and engaging in ongoing dialogue, vaccine hesitancy can be improved, ultimately protecting the health and well-being of individuals, especially those at-risk of complications.

## Figures and Tables

**Table 1 vaccines-11-01365-t001:** Assessing this Narrative Review Using the International Narrative Systematic Assessment (INSA) Tool.

Background of the Study	Vaccines have always sparked debates about its safety. The issue of vaccine hesitancy has been brought to the forefront because of COVID-19 vaccines, even more so in individuals with pre-existing skin conditions.
Objective	This review explores the key arguments surrounding vaccine hesitancy, while also addressing strategies to overcome it.
Description/ Motivation of Selection of Studies	Focus was mainly on articles published after December 2020, after the first COVID-19 vaccine was approved for emergency use.
Description of the characteristic of the included studies	Articles were identified by the PubMed search in English using keywords of COVID-19 vaccine (or SARS-CoV-2 vaccines), attitudes towards vaccines, vaccine acceptance (or acceptability or willingness or hesitancy or refusal), dermatology (or dermatological conditions or cutaneous), and vaccine fatigue.
Presentation of Results	Results of this narrative result were presented in six main sections: Vaccine Utilization Prior To COVID-19 Pandemic, Dermatology Patients and Attitude Towards COVID-19 Vaccination, Factors Affecting COVID-19 Vaccine Uptake In Dermatology Patients, Factors Affecting COVID-19 Vaccine Uptake In Dermatology Patients, Improving Vaccine Acceptance and Role of Clinicians, Healthcare Workers Towards Non-COVID-19 And COVID-19 Vaccines; and Vaccine Fatigue.
Conclusion of Narrative Review	Vaccine hesitancy within dermatological patients challenges public health efforts. A comprehensive approach, involving healthcare workers like dermatologists is crucial to tackle safety concerns, misinformation, and promote vaccine acceptance, addressing individual needs and beliefs. Maintaining vaccine acceptance, combating fatigue, and ongoing education are vital in protecting at-risk individuals during the pandemic and booster stages.
Conflict of Interests	The authors’ have declared their individual potential conflict of interests as required.

## Data Availability

Not applicable.

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
