# Peer review of "Attitudes towards COVID Vaccine and Vaccine Hesitancy in Dermatology: A Narrative Review"

_vaccines, 2023, doi:10.3390/vaccines11081365_

Round 1

Reviewer 1 Report

The authors reported the results of a review manuscript investigating the attitudes towards COVID vaccine and vaccine hesitancy in  Dermatology. The authors analyzed the causes of vaccine hesitancy and fatigue as well as reported data on other types of vaccines. Moreover, the authors reported strategies to increase vaccine acceptance underlying the role of the clinicians. Globally, the manuscript is interesting and well written and it fits wells with the aim of the Journal. I have only few suggestions.
My comments:

- Introduction: few data on the type of Covid-19 vaccines (mRNA or viral-vector based) should be reported.

- Cutaneous reactions following Covid-19 vaccination (both exacerbation and new-onset) should be discussed (doi: 10.2147/CCID.S388245).

- Strengths and limitations of the study should be discussed.

- Discussion should be implemented, reporting strategies to overcome vaccine hesitancy and highlighting the role of clinicians

Author Response

Dear reviewer,

Thank you very much for reviewing our work and comments on how to improve it:

  1. Introduction: few data on the type of Covid-19 vaccines (mRNA or viral-vector based) should be reported.

- Response: We added the types of COVID-19 vaccines as suggested, but under section 4. “Factors Affecting Covid-19 Vaccine Uptake In Dermatology Patients”

  1. Cutaneous reactions following Covid-19 vaccination (both exacerbation and new-onset) should be discussed (doi: 10.2147/CCID.S388245).

- Response: We added a short paragraph on cutaneous reaction after COVID-19 vaccinations under section 4.2.

  1. Strengths and limitations of the study should be discussed.

- Response: We added a new section 8 “Strengths and Weaknesses”

  1. Discussion should be implemented, reporting strategies to overcome vaccine hesitancy and highlighting the role of clinicians

- Response: Thank you for highlighting this. We have renamed Section 5 to “Improving Vaccine Acceptance and Role of Clinicians” to improve the readability to draw attention to this section.

Thank you very much once again for reviewing our manuscript.

Reviewer 2 Report

1-     Please provide the method

2-     Please address the process for identifying the literature search such as years considered, language, publication status, study design, and databases of coverage

3-     Please present data synthesis

4-     Please provide the practical implication of the study

Author Response

Dear reviewer,

Thank you so much for your review and comments on how to improve the manuscript. 

As our manuscript was intended to be a narrative qualitative review of the issues surrounding reasons and opinions behind vaccine hesitancy in dermatology, the scientific approach required for a systematic review would be not applicable for us. If we were to consider expanding our work to answering specific question in this area, we would definitely present the methods as per your suggestion.

Thank you very much once again for reviewing our manuscript.

Round 2

Reviewer 2 Report

Thanks to the Authors, manuscript has substantially improved, however it needs to be addressed the type of review and the method of selecting the papers

Author Response

Dear reviewer,

Thank you for your review and suggestion again! We really appreciate your taking of time for our manuscript. 

In line with your suggestion, we have added a paragraph at the end of 1. INTRODUCTION, giving a short summary of the type of review (narrative literature review) and method of selecting the paper.

Thank you very much.